

# The antioxidative properties of S-allyl cysteine not only influence somatic cells but also improve early embryo cleavage in pigs

Markéta Dvořáková[1], Ivona Heroutová[1], David Němeček[1], Kateřina Adámková[1], Tereza Krejčová[1], Jan Nevoral[1,2], Veronika Kučerová Chrpová[1], Jaroslav Petr[3] and Markéta Sedmíková[1]

[1] Department of Veterinary Sciences, Czech University of Life Sciences, Prague, Czech Republic
[2] Biomedical center, Faculty of Medicine in Pilsen, Charles University in Prague, Plzen, Czech Republic
[3] Institute of Animal Science, Prague, Czech Republic

## ABSTRACT

In vitro cultivation systems for oocytes and embryos are characterised by increased levels of reactive oxygen species (ROS), which can be balanced by the addition of suitable antioxidants. S-allyl cysteine (SAC) is a sulfur compound naturally occurring in garlic (*Allium sativum*), which is responsible for its high antioxidant properties. In this study, we demonstrated the capacity of SAC (0.1, 0.5 and 1.0 mM) to reduce levels of ROS in maturing oocytes significantly after 24 (reduced by 90.33, 82.87 and 91.62%, respectively) and 48 h (reduced by 86.35, 94.42 and 99.05%, respectively) cultivation, without leading to a disturbance of the standard course of meiotic maturation. Oocytes matured in the presence of SAC furthermore maintained reduced levels of ROS even 22 h after parthenogenic activation (reduced by 66.33, 61.64 and 57.80%, respectively). In these oocytes we also demonstrated a growth of early embryo cleavage rate (increased by 33.34, 35.00 and 35.00%, respectively). SAC may be a valuable supplement to cultivation media.

Corresponding author
Markéta Dvořáková,
dvorakova2@af.czu.cz

## INTRODUCTION

During meiotic maturation in in vitro conditions, oocytes acquire developmental competence, which is decisive with regard to the capacity of the fertilised oocyte to develop into a viable embryo (*Wassarman, 1988*). Oxidative stress negatively influences meiotic maturation by influencing the properties of its cytoskeleton. It damages the microfilaments of the maturing oocyte (*Jiao et al., 2013*), disturbs the dynamics of the microtubular network and the attachment of chromosomes to microtubules (*Choi et al., 2007*). Oxidative stress negatively influences homeostasis of calcium ions (*Ambruosi et al., 2011*). It also impairs the redistribution of cortical granules during the course of meiotic maturation, which increases the incidence of polyspermy after in vitro fertilisation (IVF) (*Jiao et al., 2013*). Excessive production of reactive oxygen species

(ROS) reduces the percentage of formed pronuclei in porcine oocytes following IVF (*Alvarez et al., 2015*).

Oxidative stress is a consequence of increased levels of ROS in cells. Balanced levels of ROS are important for the correct functioning of the organism, and are also important in the process of meiotic maturation. A proportionate amount of ROS in the follicular fluid supports germinal vesicle breakdown, by which the process of meiotic maturation begins (*Takami et al., 1999*). The follicular fluid also contains antioxidants. Balance of the levels of ROS and antioxidants in the follicular fluid is of key importance for the successful course of meiotic maturation (*Pasqualotto et al., 2004*).

In vitro cultivation systems used for the cultivation of oocytes are endangered by increased levels of ROS and the development of oxidative stress, because cultivation media contain a range of components manifesting pro-oxidative activity. These include, for example, energy sources such as lactate and pyruvate (*Hashimoto et al., 2000*), and hormones (*Markides, Roy & Liehr, 1998*). Transitory exposure of in vitro cultures to light (*Takenaka, Horiuchi & Yanagimachi, 2007*) and increased concentrations of oxygen (*Agarwal, Saleh & Bedaiwy, 2003*) also increases the production of ROS in cultivation systems.

Balancing increased levels of ROS in a cultivation medium by the addition of suitable antioxidants may prevent the development of oxidative stress and thus have a positive influence on early embryo cleavage of matured oocytes. Several sources of antioxidant substances are known, and more are being sought within the framework of ongoing studies. These, for example, include the amino acid cysteine, which reduces levels of ROS in maturing bovine oocytes (*Morado et al., 2009*). A cysteine derivative, N-acetyl cysteine (NAC), positively influences the formation of pronuclei and the development of blastocytes in vitro in pigs (*Whitaker, Casey & Taupier, 2012*).

Antioxidants also include a further cysteine derivative, the sulfur compound S-allyl cysteine (SAC), which is responsible for the high antioxidant activity of garlic (*Colín-González et al., 2015*). SAC is known for its anti-apoptotic and antioxidant effects in a range of types of somatic cells. SAC manifests antioxidant properties for example in the nervous (*Tsai et al., 2011*) and cardiovascular systems (*Louis et al., 2012*). *Takemura et al. (2014)* published a study demonstrating the antioxidant effects of SAC on rat sperm.

In somatic cells SAC manifests better antioxidant properties in comparison with cysteine. Upon oral administration to mice it brought about a larger increase in the activity of antioxidant enzymes in plasma, the kidneys and liver in comparison with cysteine (*Hsu et al., 2004*). In addition, according to *Dion, Agler & Milner (1997)*, SAC is more effective than cysteine in the protection of liver cells against the mutagenic effects of nitrosomorpholine. To date no study has been published dealing with the potential antioxidant effects of SAC on maturing oocytes.

The aim of the presented study was to test the hypothesis that SAC influences meiotic maturation of porcine oocytes and early embryo cleavage during in vitro cultivation.

## MATERIALS AND METHODS

### Collection and cultivation of oocytes and evaluation of meiotic maturation

Oocytes were obtained from ovaries through aspiration from follicles (2–5 mM in diameter) with 20G needles and cultured in a modified M199 medium (Gibco BRL, Life Technologies, Carlsbad, CA, USA) supplemented with calcium L-lactate (2.75 mM; Sigma Aldrich, USA), sodium pyruvate (0.25 mg/mL; Sigma Aldrich, USA), gentamicin (0.025 mg/mL; Sigma Aldrich, USA), HEPES (6.3 mM; Sigma Aldrich, USA), 10% (v/v) foetal calf serum (Gibco BRL, Life Technologies, Germany), albumin (5 mg/mL; Sigma Aldrich, USA) and 13.5 IU eCG: 6.6 IU hCG/mL (P.G. 600, Intravet, Boxmeer, Netherlands). Oocytes were cultured with SAC (Sigma Aldrich, USA) in concentrations of 0.0 (control), 0.1, 0.5, 1.0 and 5.0 mM for 24 and 48 h (39 °C; 5% $CO_2$). The concentration of 5.0 mM was applied in experiments concerning nuclear maturation only.

After culture oocytes were denuded of cumulus cells by repeated pipetting through a narrow glass capillary and mounted on slides. The following stages of meiotic maturation were evaluated under a phase contrast microscope: germinal vesicle (GV), metaphase I (MI) and metaphase II (MII).

### MPF/MAPK double assay

Kinase Double Assay was performed according to *Kubelka et al. (2000)*. Briefly, samples were prepared from 15 oocytes cultivated with SAC by 5 µl extraction buffer addition and immediately frozen (−80 °C). Specific substrates H1 (Histone H1) and Myelin Basic Protein (MBP) were phosphorylated using radioactive labelled $[\gamma\text{-}^{32}P]ATP$, 500 µCi/mL (GE Healthcare Life Sciences, USA) and separated by SDS-PAGE. The signal intensities were measured by IP-plate, FLA 7000 reader (GE Healthcare Life Sciences, USA) and Multi-Gauge 2.0 software (Fujifilm, Japan). The obtained data was expressed relative to MPF/MAPK activities in oocytes in GV stage where we expect the lowest measured activities of MPF and MAPK.

### Measurement of hyaluronic acid production within cumulus-oocyte complexes

Groups of 25 cumulus-oocyte complexes (COCs) were cultured for 24 and 48 h, washed four times in 500 µl PBS-PVA (0.01%) transfering them gently using a 50 µl pipette. Oocytes were denuded from cumulus cells and removed from samples. Samples were transferred into Eppendorf tubes, enzymatically digested using lyase from *Streptomyces hyalurolyticus* (20 µl/mL; Sigma-Aldrich, USA) at 39 °C overnight, centrifuged (5 min; 10,000 rpm, 4 °C) and measured in a Helios Epsilon spectrophotometer (Verkon, Czech Republic) at 216 nm.

### Reactive oxygen species measurement

Reactive oxygen species production was evaluated in oocytes after 24 and 48 h of meiotic maturation and zygotes after 22 h of cultivation. Oocytes and zygotes were stained with 10 µM 2′,7′-dichlorodihydrofluorescin diacetate (Sigma-Aldrich, USA) (20 min; 39 °C) and mounted on glass. Samples were evaluated using a confocal microscope

(Leica SPE) and NIS Elements 4.0 software (Laboratory Imaging, Czech Republic). The results were expressed as the relative fluorescence intensity and related to the control group.

## Parthenogenic activation of oocytes

Parthenogenic activation was carried out according to *Jílek et al. (2001)*. Briefly, matured oocytes denuded from cumulus cells were activated using calcium ionophore A23187 (25 μM, 5 min; Sigma Aldrich, USA) and 6-dimethylaminopurine—6-DMAP (2 mM, 2 h; Sigma Aldrich, USA) and cultivated in a modified M199 medium without hormones for 22 h. Activating potential was evaluated as the ratio of zygotes with 1 or 2 pronuclei and cleaving embryos. Early embryo cleavage was evaluated as the ratio of cleaving embryos among activated oocytes.

## Statistical analysis

Each experimental group contained 120 oocytes for nuclear maturation and parthenogenic activation assessment, 100 for hyaluronic acid production assessment and 60 for MPF/MAPK Double Assay and ROS measurement. All experiments were repeated four times. SAS 9.0 Software (SAS Institute Inc., Cary, North Carolina, USA) was used for the statistical analyses. Significant differences between groups were determined using the one-way ANOVA test followed by Scheffe's method. $P < 0.05$ were considered significant. Statistically significant differences among different groups of oocytes are indicated by different superscripts.

## Design of the experiments

Experiment 1 was performed in order to investigate the effect of SAC on the meiotic maturation of porcine oocytes. The oocytes were cultured for 24 and 48 h in vitro in the maturation medium described above, and supplemented with SAC in different concentrations. At the end of culture, stages of meiotic maturation, MPF and MAPK activity and hyaluronic acid production were evaluated.

Experiment 2 was focused on the effect of SAC on ROS levels in oocytes after 24 and 48 h of meiotic maturation and zygotes after subsequent 22 h of cultivation. We investigated the effect of SAC on ROS production as an indicator of oxidative stress and therefore quality of oocytes.

Experiment 3 was performed in order to evaluate the effect of SAC applied during meiotic maturation on subsequent parthenogenic activation. Activating potential and early embryo cleavage were considered as indicators of oocyte quality.

# RESULTS

## Effect of S-allyl cysteine on meiotic maturation of porcine oocytes

Nuclear maturation, cytoplasmic maturation and hyaluronic acid production were used as markers of successful meiotic maturation. Nuclear maturation was evaluated as a stage of meiotic maturation. SAC did not influence nuclear maturation despite the concentration of 5 mM which disrupts the standard course of the process (see Fig. 1A). This concentration was not applied in further experiments.

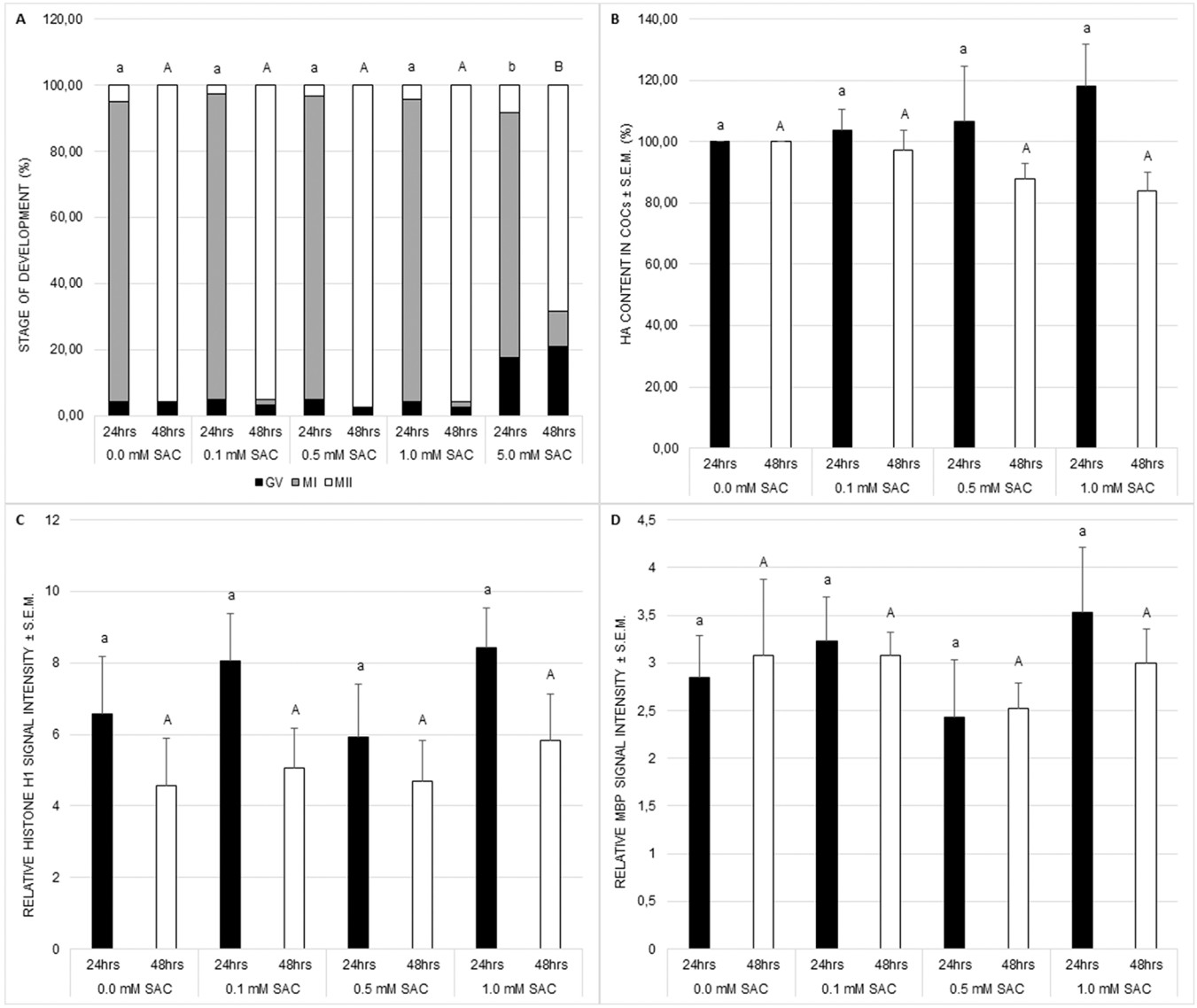

**Figure 1 Effects of various SAC concentrations on porcine oocyte meiotic maturation after 24 and 48 h of cultivation.** (A) Effects of SAC on nuclear maturation. GV–oocytes in the germinal vesicle stage, MI–oocytes in metaphase I, and MII–oocytes in metaphase II. Statistically significant differences between nuclear maturation stages (GV, MI, and MII) from various SAC concentrations are indicated by different superscripts: a, b–differences between nuclear maturation stages from various SAC concentrations after 24 h of cultivation ($P < 0.05$). A, B–differences between nuclear maturation stages from various SAC concentrations after 48 h of cultivation ($P < 0.05$). Data are presented as a mean of four replicates (n = 120 in each group). (B) Effects of SAC on hyaluronic acid (HA) content within COCs. Statistically significant differences between HA contents from various SAC concentrations are indicated by different superscripts: a, b–differences between HA contents from various SAC concentrations after 24 h of cultivation ($P < 0.05$). A, B–differences between HA contents from various SAC concentrations after 48 h of cultivation ($P < 0.05$). Data are presented as a mean ± S.E.M. of four replicates (n = 100 in each group). (C) Effects of SAC on MPF activity. Phosphorylated histone H1 signal intensity is related to signal intensity in GV oocytes and reflects changes in MPF activity. Statistically significant differences between relative histone H1 signal intensities from various SAC concentrations are indicated by different superscripts: a, b–differences between relative histone H1 signal intensities from various SAC concentrations after 24 h of cultivation ($P < 0.05$). A, B–differences between relative histone H1 signal intensities from various SAC concentrations after 48 h of cultivation ($P < 0.05$). Data are presented as a mean ± S.E.M. of four replicates (n = 60 in each group). (D) Effects of SAC on MAPK activity. Phosphorylated MBP signal intensity is related to signal intensity in GV oocytes and reflects changes in MAPK activity. Statistically significant differences between relative MBP signal intensities from various SAC concentrations are indicated by different superscripts: a, b–differences between relative MBP signal intensities from various SAC concentrations after 24 h of cultivation ($P < 0.05$). A, B–differences between relative MBP signal intensities from various SAC concentrations after 48 h of cultivation ($P < 0.05$). Data are presented as a mean ± S.E.M. of four replicates (n = 60 in each group).

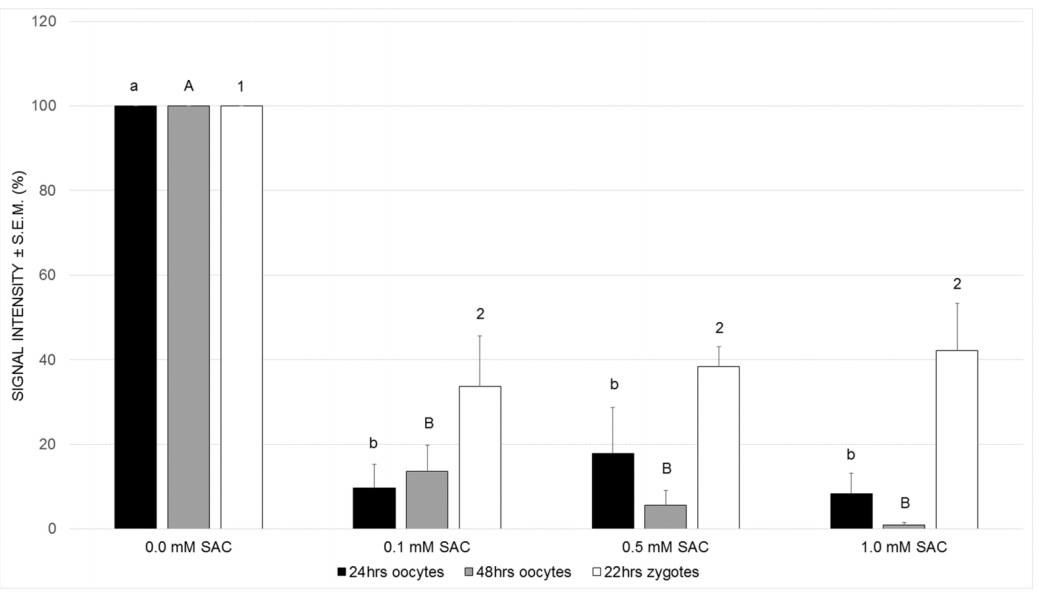

**Figure 2 Effects of SAC on ROS production in porcine oocytes after 24 and 48 h of cultivation, and in zygotes after 22 h of cultivation.** Statistically significant differences between ROS levels from various SAC concentrations are indicated by different superscripts: a, b–differences between ROS levels from various SAC concentrations after 24 h of cultivation ($P < 0.05$). A, B–differences between ROS levels from various SAC concentrations after 48 h of cultivation ($P < 0.05$). 1, 2–differences between ROS levels from various SAC concentrations after 22 h of cultivation of parthenogenetically activated oocytes ($P < 0.05$). Data are presented as a mean ± S.E.M. of four replicates ($n = 60$ in each group).

Cytoplasmic maturation was evaluated as MPF and MAPK activities. MPF and MAPK activities as well as hyaluronic acid production by COCs were not influenced by SAC (see Figs. 1B–1D).

## Effect of S-allyl cysteine on reactive oxygen species production in porcine oocytes and zygotes

In these experiments we measured levels of ROS in order to prove our hypothesis that SAC has antioxidant activity in oocytes, as has been proven in somatic cells.

Primarily, we cultivated maturing oocytes in the presence of SAC in concentrations 0.1, 0.5 and 1.0 mM and evaluated levels of ROS within oocytes after 24 and 48 h of cultivation. We observed a significant decrease in ROS production in all experimental groups after 24 as well as 48 h of cultivation (see Fig. 2).

Obtaining these results, we continued in experiments by parthenogenic activation of oocytes matured in the presence of SAC. We evaluated ROS levels in zygotes 22 h after parthenogenic activation. According to our results, parthenogenetically activated zygotes maintained their antioxidant capacity and exhibited lowered ROS levels when compared to the control group (see Fig. 2).

## Effect of S-allyl cysteine on oocyte parthenogenic activation

In the following experiments we focused on the effect of SAC on activating potential and early embryonic cleavage as indicators of embryo quality. SAC in concentrations

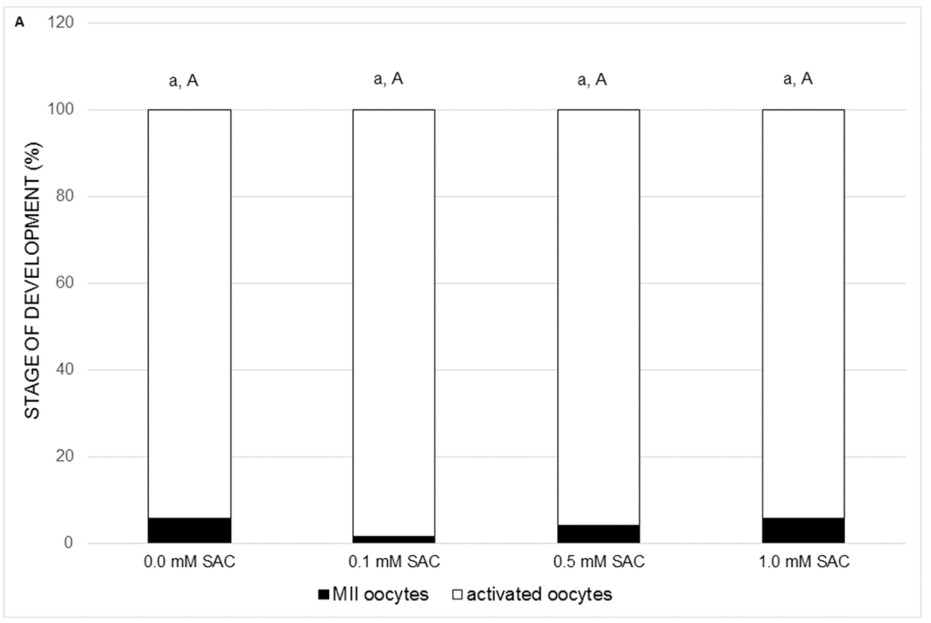

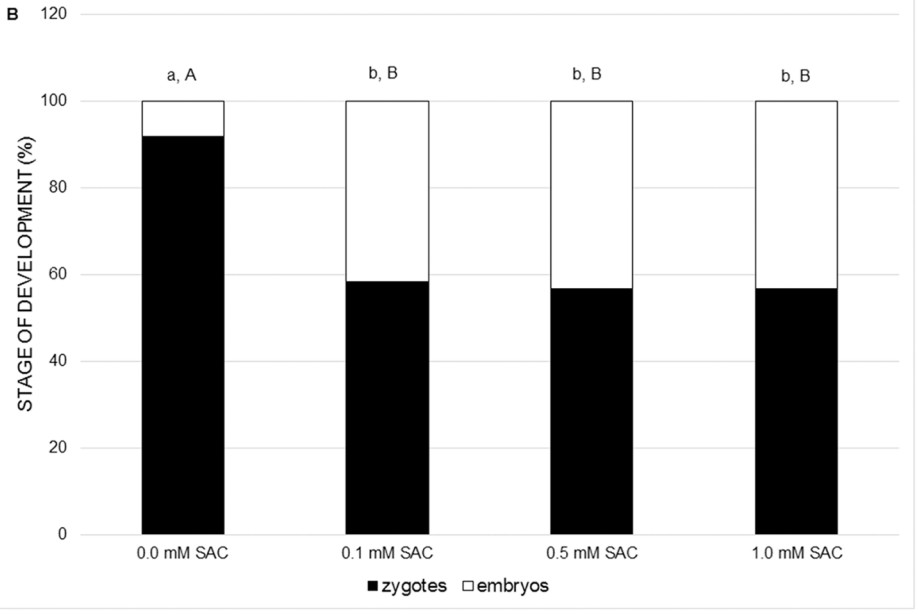

**Figure 3 Effects of various SAC concentrations on parthenogenic activation of porcine oocytes.**
(A) Effects of SAC on the activating potential of parthenogenetically activated oocytes after 22 h of cultivation. MII oocytes–oocytes in metaphase II, activated oocytes–zygotes with one or two pronuclei and 2–3-cell cleaving embryos. Statistically significant differences between oocyte developmental stages from various SAC concentrations are indicated by different superscripts: a, b–differences between percentages of MII oocytes from various SAC concentrations ($P < 0.05$). A, B–differences between percentages of activated oocytes from various SAC concentrations ($P < 0.05$). Data are presented as a mean of four replicates (n = 120 in each group). (B) Effects of SAC on the early embryo development of parthenogenetically activated oocytes after 22 h of cultivation. Zygotes had one or two pronuclei; embryos consisted of 2 or 3 cells. Statistically significant differences between oocyte developmental stages from various SAC concentrations are indicated by different superscripts: a, b–differences between percentages of zygotes from various SAC concentrations ($P < 0.05$). A, B–differences between percentages of embryos from various SAC concentrations ($P < 0.05$). Data are presented as a mean of four replicates (n = 120 in each group).

0.1, 0.5 and 1.0 mM did not affect activating potential, however it enhances early embryo cleavage in comparison to the control (see Figs. 3A and 3B).

## DISCUSSION

In our study, we demonstrated that SAC reduces levels of ROS in porcine oocytes during their maturation in vitro. In the case of oocytes maturing in the presence of SAC we did not observe deviations during the course of nuclear maturation or in the activity of kinases of key importance for the meiotic maturation of oocytes. Expansion of cumulus was also not influenced by cultivation. After parthenogenic activation we observed a higher proportion of cleaving embryos in oocytes maturing in the presence of SAC. The capacity of SAC to reduce intracellular levels of ROS has been described in somatic cells (*Tsai et al., 2011*). To the best of our knowledge, our study is the first to describe this effect of SAC on in vitro maturing mammal oocytes.

The marked reduction of intracellular levels of ROS observed in our study in porcine oocytes maturing in vitro in a medium enriched with SAC can be explained by the fact that both cysteine and the allyl group have antioxidant properties (*Chung, 2006*). It is known that the addition of cysteine alone or its derivatives (e.g. NAC) to the cultivation medium is capable of achieving a suppression of intracellular levels of ROS upon cultivation of oocytes and embryos in vitro (*Alvarez et al., 2015*; *Giorgi et al., 2015*). According to several in vivo experiments SAC has stronger antioxidant effects on various types of tissues than cysteine alone (*Hsu et al., 2004*) or than NAC (*Mizuguchi et al., 2006*).

In our study, the reduction of ROS levels did not have a significant impact on the observed aspects of maturation of porcine oocytes. This could indicate that porcine oocytes are relatively resistant to the effects of ROS. This is attested to also by the observations of *Alvarez et al. (2015)*, in which the increase of ROS levels had no impact on maturation. However, *Alvarez et al. (2015)* describe an increase in the proportion of oocytes maturing to metaphase II after a reduction of ROS by the addition of cysteine to the cultivation medium for maturation.

In our experiments we did not demonstrate the influence of SAC added during meiotic maturation on the proportion of oocytes emerging from metaphase II following parthenogenetic activation (thus the effect on the activation rate). However, in all applied concentrations (0.1, 0.5 and 1.0 mM), SAC increased the percentage of cleaving zygotes following parthenogenetic activation. A similar effect has also been described in the case of cysteine which, in the study by *Li et al. (2014)*, increased early embryo cleavage of porcine oocytes following ICSI, and also in the case of NAC, which improved the formation of male pronuclei and subsequent embryonic development (*Whitaker, Casey & Taupier, 2012*). On the basis of our results it is possible to conclude that SAC positively influences early embryo cleavage, a significant indicator of the quality of activated oocytes. This effect may be the result of suppression of ROS levels in zygotes, which persists from previous maturation of oocytes in the presence of SAC.

SAC need not act on oocytes cultivated in vitro only as an antioxidant reducing intracellular levels of ROS, but may also have an indirect effect via other target systems. SAC is also capable of increasing the activity of antioxidant enzymes such as catalase

and glutathione peroxidase (*Hsu et al., 2004*), by increasing intracellular levels of glutathione, which is known as a significant antioxidant responsible for uptake of ROS in cells (*Kohen & Nyska, 2002*). Also significant may be the capacity of SAC to increase the intracellular concentration of hydrogen sulfide (*Szabó, 2007*), which tanks among significant gaseous signalling molecules termed gasotransmitters (*Kamoun, 2004*). Hydrogen sulfide plays a significant role in regulating the maturation of mammal oocytes (*Nevoral et al., 2014*).

The extent to which hydrogen sulfide contributed to the effects of SAC we observed is not clear. In this study, in the case of COCs, after cultivation with SAC we did not observe an acceleration of maturation of oocytes or a suppression of expansion of cumulus cells, which is manifested under the influence of hydrogen sulfide on COCs (*Nevoral et al., 2014*). On the other hand, sulfide ions may have a whole range of indirect effects on oocytes. Hydrogen sulfide influences the activity of several proteins, including enzymes and the ion channels of their sulfhydration (*Paul & Snyder, 2012*). Sulfide ions also have an effect on the activity of other gasotransmitters—nitric oxide and carbon monoxide (*Li, Hsu & Moore, 2009*), which may significantly influence the maturation of oocytes (*Jablonka-Shariff & Olson, 1998*).

## CONCLUSIONS

Further experiments will be required for a more detailed clarification of the effect of SAC on oocytes and their developmental competence. Our experiments indicate that SAC is an antioxidant suitable as a supplement to cultivation media for oocytes because it does not disturb the course of meiotic maturation, which is sensitive to imbalance of ROS. The addition of SAC to in vitro cultivation systems may make a significant contribution to the success of in vitro maturation and subsequent activation and the early embryonic development of oocytes.

### Funding

This work was supported by the Ministry of Agriculture of the Czech Republic (NAZV–Project No. QJ1510138; MZeRO 0714) and by Internal Grant Agency of the Czech University of Life Sciences Prague (CIGA) (Project No. CZU20142049). The funders had no role in study design, data collection and analysis, decision to publish, or preparation of the manuscript.

### Grant Disclosures

The following grant information was disclosed by the authors:
Ministry of Agriculture of the Czech Republic: NAZV–QJ1510138; MZeRO 0714.
Czech University of Life Sciences Prague (CIGA): CZU20142049.

### Competing Interests

The authors declare that they have no competing interests.

## Author Contributions

- Markéta Dvořáková conceived and designed the experiments, performed the experiments, analyzed the data, wrote the paper, prepared figures and/or tables.
- Ivona Heroutová performed the experiments.
- David Němeček performed the experiments.
- Kateřina Adámková performed the experiments.
- Tereza Krejčová performed the experiments.
- Jan Nevoral performed the experiments.
- Veronika Kučerová Chrpová performed the experiments.
- Jaroslav Petr conceived and designed the experiments, reviewed drafts of the paper.
- Markéta Sedmíková conceived and designed the experiments, reviewed drafts of the paper.

## Data Deposition

The raw data has been supplied as Supplemental Dataset Files.

## Supplemental Information

Supplemental information for this article can be found online at http://dx.doi.org/10.7717/peerj.2280#supplemental-information.

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
