# Peer review of "The antioxidative properties of S-allyl cysteine not only influence somatic cells but also improve early embryo cleavage in pigs"

_PeerJ, doi:10.7717/peerj.2280_

## Round 0.1 · original submission · Major Revisions

Please address the issues raised by the two reviewers.

Reviewer 1 ·

Basic reporting

No comments

Experimental design

The experimental design is simple and clear.

Validity of the findings

The results are clear and well supported.

Additional comments

The authors demonstrated that SAC is very effective in reducing ROS levels in porcine oocytes during and after their maturation in vitro, and there is a higher proportion of early cleaving embryos when the culture medium was supplemented with SAC. While with these positive effects, addition of SAC didn’t affect the meiotic process and cumulus expansion, implicating the potential of adopting SAC as a new anti-oxidant additive for in vitro culturing of oocytes. The experimental designs are sound and the conclusions that SAC reduces ROS levels in maturing oocytes and increases early embryo cleavage rate are well supported.

However, the authors failed to make a direct comparison between SAC and other known anti-oxidants such as cysteine to compare their abilities in reducing ROS, which could make the specific role of SAC stand out of other anti-oxidants. Although from previous studies it is true that in somatic cells SAC is better than cysteine in reducing ROS.

Other points:
-The title may be changed to more accurately reflect the conclusion that SAC (instead of using “sulfur garlic”) could reduce ROS levels and increase early embryo cleavage (instead of using the statement “oocyte viability”).
-The authors only recorded early embryo cleavages 22 hours after parthenogenic activation, and claimed SAC is good for oocyte viability. Although increased viability could be achieved through reducing oxidative stress in oocytes, the authors didn’t check the viability itself directly.
-The experiments and conclusions are simple and clear, but the discussion part is too long and a lot of the texts is not relevant to authors’ main conclusion. For example, for the discussion of ROS balance is important for oocytes, this is well known and not necessary to discussed too much.
-Line 52, IVF abbreviation better explained when first used.
-Line 93 and 105, the texts are redundant.
-Line 177, itclics
-Line 210, “the effects of SAC are evidently stronger…. This could also have been the case in our study, in which we did not compare the effects of SAC with the effects of cysteine and …”. Here without actual data, it is not able to conclude “this could also have been the case in our study”.

The effects of SAC on oocytes worth further study. The paper could be accepted after revision, providing they modify the texts and make it clearer and more precise, and tone down some of the statements.

·

Basic reporting

The authors demonstrated clearly how their work fits into current issues present in oocyte and zygote culture system.

Experimental design

The experimental design is clearly and supports the two aims of the study.

Validity of the findings

The authors did a lot of work. Each figure panel has observed more than 100 cells, and the experiments were repeated four times. These make the conclusions more powerful.

Additional comments

In this study, the authors did a nice job on evaluating SAC as an antioxidant reagent on maturing oocytes. SAC is able to reduce ROS level in maturing oocytes while has no disruption of meiotic maturation. Moreover, the quality of in vitro matured oocytes is improved with SAC treatment. This study demonstrates a potential application of SAC in IVF.
Minor points
1. In Figures 1-3, statistically significant differences are indicated by different superscripts. The authors should indicate what those superscripts represent. For example, a, p<0.05; b, p<0.01.
2. All the data are shown as mean±SEM, but some of the panels did not show SEM. They are Figure 1A, and Figure 3.
3. For the statistical analysis, the author indicated significant differences were determined using the ANOVA test. Which type of ANOVA was used?
4. In Figure 1A, what are the meanings of b1A1ß1, b2B2a2?
5. For experimental design, why choosing 24 and 48 hrs for oocytes and 22 hrs for zygotes?
6. In Figure 2, SAC treatment seems to have stronger effect of reducing ROS level on oocytes than on zygotes. What might be the reason?
7. For ROS level, what is the percentage of ROS usually found in those oocytes and zygotes without treatment of SAC? Is it reaching a level that is harmful?
8. line 198, “in vivo” should be “in vitro”?
9. line 201, SAC treatment results in more cleaving embryos in oocytes maturing. What is the safe yet efficient proportion of cleaving embryos?

---

## Round 0.2 · accepted · Accept

Thanks for making the revision.